# Exploiting Hierarchical Taxonomies in Prompt-based Continual Learning

## Abstract

Drawing inspiration from human learning behaviors, this work proposes a novel approach to mitigate catastrophic forgetting in Prompt-based Continual Learning models by exploiting the relationships between continuously emerging class data. We find that applying human habits of organizing and connecting information can serve as an efficient strategy when training deep learning models. Specifically, by building a hierarchical tree structure based on the expanding set of labels, we gain fresh insights into the data, identifying groups of similar classes could easily cause confusion. Additionally, we delve deeper into the hidden connections between classes by exploring the original pretrained model's behavior through an optimal transport-based approach. From these insights, we propose a novel regularization loss function that encourages models to focus more on challenging knowledge areas, thereby enhancing overall performance. Experimentally, our method demonstrated significant superiority over the most robust state-of-the-art models on various benchmarks. Our code is available at https://anonymous.4open.science/r/HierC-089B/.

## 1 Introduction

Continual Learning (CL) (Wang et al., 2024; Lopez-Paz & Ranzato, 2017) is a research direction focused on realizing the human dream of creating truly intelligent systems, where machines can learn on the fly, accumulate knowledge, and operate in constantly changing environments as a human's companion. Despite the impressive capabilities of A.I systems, continual learning remains a challenging scenario due to the tendency to forget obtained knowledge when facing new ones, known as *catastrophic forgetting* (French, 1999). In dealing with this challenge, traditional CL methods often rely on storing past data for replaying during new tasks, which can raise concerns about memory usage and privacy. To overcome this limitation, recent methods proposed leveraging the generalizability of pre-trained models (Han et al., 2021; Jia et al., 2022) as frozen backbones to solve sequences of CL tasks (Wang et al., 2022c; Smith et al., 2023; Li et al., 2024).

While these pre-trained-based methods have demonstrably achieved impressive results, only consider forgetting caused by changes in learned prompts or differences between the (prompt-based) models chosen for training and testing (Wang et al., 2023; Tran et al., 2023; Zhanxin Gao, 2024). Further completing those arguments, we show that forgetting of old knowledge also comes from the uncontrolled growth of new classes in the latent space. That is, models are confused in distinguishing between old and new classes, which many methods overlook when training tasks independently (Smith et al., 2023; Wang et al., 2022d). Furthermore, we find that current approaches only utilize limited information from the training dataset and treat class labels equally during training, resulting in missing opportunities to further enhance model representations and mitigate forgetting more effectively.

In addition, we find that human natural learning behavior has many valuable aspects, especially the habit of analyzing data, organizing them in a meaningful way, and finding connections between old and new knowledge (Schön, 1983; Bransford et al., 2000; Sweller, 1988; Mayer, 2005), thereby improving the ability to understand, remember, and reproduce information. Inspired by this, we investigate the characteristics of current common benchmark datasets as well as the behavior of pre-trained-based CL models, showing that the incoming data classes over time can always be categorized into consistent groups. Each such group usually includes class data with similar semantic

characteristics, which is easy to get confused by models and should be paid special attention to during training.

Therefore, we propose a training strategy that constantly considers emerging data in groups, following a hierarchical tree-like taxonomy developed based on expert/domain knowledge. During training new tasks, the model references information from old classes in the tree. Especially, the feature extractor is encouraged to maximally contrast and distinguish concepts/labels within the same group, promoting the learning of common features that can be transferred to new concepts/labels in the same group in future tasks. This strategy not only mitigates forgetting when new classes emerge but also consolidates domain-specific knowledge. Furthermore, we observe that images belonging to concepts/labels within the same group in the hierarchical taxonomy share strong visual and semantic correlations, leading to overlapping representations in the latent space, which compromises performance. By encouraging the feature extractor to separate and contrast the representations of images in these concepts/labels more distinctly, we effectively reduce the overlap of easily confused classes, thereby improving performance.

**Contribution.** We name our method as *Exploiting Hierarchical **T**axonomies in Prompt-based Continual Learning* (TCL), and summarize our main contributions as follows:

- We introduce a new perspective to explain the reason for catastrophic forgetting in pretrained-based Continual learning models, which potentially comes from the uncontrolled growth of incoming classes on the latent space.
- Originating from the research findings of Cognitive Science, we propose a novel approach to reduce forgetting by exploiting relationships between data. By dynamically building label-based hierarchical taxonomies and leveraging the initial behavior of the pretrained model, we can identify the challenging knowledge areas that further focus, and determine how to learn them during the sequence of tasks. Based on this taxonomical structure, our testing strategy further improves model performance.
- We empirically evaluate the effectiveness of our method against current state-of-the-art pre-trained-based baselines across various benchmarks.

**Organization.** The rest of the paper is structured as follows. In Section 2, we present related work. Then in Section 3, we formulate the problem and introduce a new perspective to explain the cause of forgetting in CL models. Section 4 transitions from the motivation provided by Cognitive Science insights to the proposed training and testing strategy, emphasizing the importance of exploiting relationships between class data. Section 5 presents the experimental results, and finally, we discuss the limitations and suggest future directions in Section 6.

## 2 RELATED WORK

**Class Incremental Learning.** This is one of the most challenging and widely studied CL scenarios (Van de Ven & Tolias, 2019; Wang et al., 2023), where task identity is unknown during testing, and data of previous data is inaccessible during current training (Masana et al., 2023; Rebuffi et al., 2017; Hou et al., 2019; Guo et al., 2022). This work follows the setting of CIL and proposes a novel approach to mitigate forgetting and improve performance for prompt-based CL models.

**Prompt-based Continual Learning.** This line of work exploits the power of pre-trained backbone to quickly adapt to the sequence of downstream tasks by updating just a small number of parameters (prompts). Initial work like Wang et al. (2022d;c); Smith et al. (2023) typically assign a set of prompts to tasks, enhancing the adaptability of the backbone to downstream tasks. However, the absence of explicit constraints can lead to feature overlapping between classes from different tasks. Therefore, recent methods employ some types of contrastive loss (Wang et al., 2023; Li et al., 2023) or utilize Vision Language models (Wang et al., 2022a; Nicolas et al., 2024) to better separate features from tasks. However, they treat all classes equally during training, missing the opportunity to learn in challenging areas where classes have many similarities and are easily confused. In this work, we propose a novel approach to exploit the relationships within data, allowing the model to recognize groups of these classes, and develop a deeper understanding of the respective knowledge areas, thereby reducing forgetting and enhancing its ability to learn new tasks.

## 3 CLASS INCREMENTAL LEARNING AND FORGETTING IN PROMPT-BASED MODELS

### 3.1 PROBLEM FORMULATION AND NOTATIONS

We consider the Class Incremental Learning setting (Zhou et al., 2024; Lopez-Paz & Ranzato, 2017; Wang et al., 2023), where a model has to learn from a sequence of $T$ visual classification tasks without revisiting old task data during training or accessing task IDs during inference. Each task $t \in \{1, ..., T\}$ has a respective dataset $\mathcal{D}_t$, containing $n_t$ i.i.d. samples $(\boldsymbol{x}_t^i, y_t^i)_{i=1}^{n_t}$. Let $D_c = (X_c, Y_c)$ denote the data corresponding to the class label $c$.

In this work, we design our model as a composition of two components: *a pre-trained ViT backbone* $f_\Phi$ and *a classification head* $h_\psi$. That is, we have the model parameters $\theta = (\Phi, \psi)$. Similar to other existing prompt-based methods, we incorporate into the pre-trained ViT a set of prompts $\boldsymbol{P}$. We denote the overall network after incorporating the prompts as $f_{\Phi, \boldsymbol{P}}$.

### 3.2 FORGETTING IN PROMPT-BASED CONTINUAL LEARNING

In CL models, changes in the dataset, including inputs $\boldsymbol{X}$ in the input space $\mathcal{X}$ and labels $\boldsymbol{Y}$ in the label space, $\mathcal{Y}$ lead to the changes of model's behavior *(feature shift)* and thus model's performance on previously learned tasks to decrease significantly (i.e., *catastrophic forgetting*). Current prompt-based CL methods, which leverage the power of pretrained models, attribute forgetting/feature shift either to **(I)** changes in parameters from the backbone when using the common prompt pool $\boldsymbol{P}$ for all tasks (Wang et al., 2022d;c) or to **(II)** the inherent mismatch between the models used at training and testing. Specifically, let $\hat{t}(\boldsymbol{x})$ and $t(\boldsymbol{x})$ as the chosen promptID and the ground-truth promptID for $\boldsymbol{x}$, respectively. We may have $f_{\hat{\theta}_t} = f_{\Phi, \boldsymbol{P}_{\hat{t}}} \neq f_{\theta_t} = f_{\Phi, \boldsymbol{P}_t}$ because there are chances that the promptID $\hat{t} \neq t$, where $\hat{t}$ is predicted by pretrained backbone $f_\Phi$ (Figure 1a), as discussed and analyzed in Zhanxin Gao (2024); Tran et al. (2023). To complement these views, below we provide an empirical study to offer new insights about the reason for forgetting in this type of model, which arises from overlapping between old and new class representations.

Firstly, to completely eliminate concerns about changing learned parameters, we consider methods that propose using a distinct set of prompts $\boldsymbol{P}_t$ to a specific task $t$. Then, the remaining potential factor of *forgetting by feature shift* is the difference between the prompt chosen at inference time and the one used during training (i.e., $\boldsymbol{P}_t \neq \boldsymbol{P}_{\hat{t}}$). Thus, we conduct experiments on HiDE (Wang et al., 2023) (i.e., the latest SOTA in prompt-based CL) to measure the differences between the features formed when using these two prompts. Particularly, we consider $\boldsymbol{x} \in D_1$ that belongs to the first task and measure the L2 Wasserstein distance $W_2(Q, \hat{Q})$ (Kantorovich, 1939) between $Q$ (i.e., the latent distribution corresponding to $\boldsymbol{P}_t$, which consists of $f_{\Phi, \boldsymbol{P}_{t(\boldsymbol{x})}}(\boldsymbol{x})$ for $\boldsymbol{x} \sim D_1$) and $\hat{Q}$ (i.e., the latent distribution corresponding $\boldsymbol{P}_{\hat{t}}$ after learning the last task, consists of $f_{\Phi, \boldsymbol{P}_{\hat{t}(\boldsymbol{x})}}(\boldsymbol{x})$ for for $\boldsymbol{x} \sim D_1$). The results in Table 1 show that the difference of these distributions is apparently negligible in many cases.

For a closer look, besides *the main classification head* $h_\psi$ used for all classes so far, at the end of each task $t$, we set up *a specific classifier* $s_t$ optimized on the frozen latent space of $D_t$ and then kept fixed. From now on, we refer to the accuracy measured on $D_t$ using $s_t$ as *'within task accuracy'*, and the accuracy using $h_\psi$ as *'true accuracy'* of this task. The results in Figure 1b, show that *within task accuracy* of the first task stays almost unchanged, which concurs with our observation on the negligible shift between $f_{\Phi, \boldsymbol{P}_{\hat{t}(\boldsymbol{x})}}(\boldsymbol{x})$ and $f_{\Phi, \boldsymbol{P}_{t(\boldsymbol{x})}}(\boldsymbol{x})$. Meanwhile, we observe a significant decrease in the corresponding *true performance* in Figure 1c, raising the question of whether we have overlooked additional factors contributing to final forgetfulness (Figure 1d), beyond the issue of selecting incorrect task prompts during inference.

Considering the *inference feature space* of $f_{\Phi, \boldsymbol{P}_{\hat{t}(\boldsymbol{x})}}(\boldsymbol{x})$, we can see that as more tasks arrive, the number of classes increase, making the space fuller and increasing the possibility of overlap between class distributions. To demonstrate this point, we provide t-SNE visualization of class representations in Figure 4 and the respective illustration in Figure 5. In particular, after Task 1, we have representations of "oak tree", "mouse" and "porcupine" located in quite separate locations. However, when Task 2 and then Task 3 arrive, the appearance of "willow tree" and "pine tree" makes the

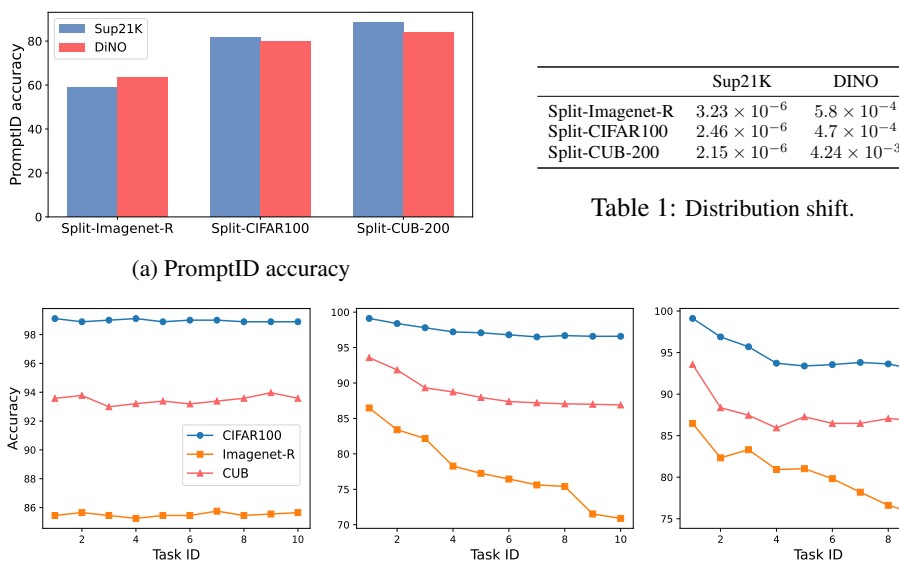

(a) PromptID accuracy

Table 1: Distribution shift.

|  | Sup21K | DINO |
|---|---|---|
| Split-Imagenet-R | $3.23 \times 10^{-6}$ | $5.8 \times 10^{-4}$ |
| Split-CIFAR100 | $2.46 \times 10^{-6}$ | $4.7 \times 10^{-4}$ |
| Split-CUB-200 | $2.15 \times 10^{-6}$ | $4.24 \times 10^{-3}$ |

(b) Within task accuracy on $D_1$    (c) True accuracy on $D_1$    (d) Overall accuracy

Figure 2: **Empirical study about forgetting (HiDE). (a)** Average accuracy of promptID prediction for all tasks; **(b)** Accuracy of the first task over time, using classification head $s_1$; **(c)** Accuracy of the first task over time, using classification head $h_\psi$; **(d)** Average accuracy on all tasks so far, after learning each task. **Table 1** (Distribution shift) reports L2 Wasserstein distance between the latent distributions corresponding to $P_t$ and $P_{\hat{t}}$ of data task 1, after learning the final task.

latent space become fuller, and "oak tree" no longer maintains the separation from the remaining classes as before and even its representation may even be misassigned to other classes, leading to a remarkable drop in performance.

Therefore, another key cause of catastrophic forgetting in prompt-based continual learning (CL) that should be recognized is *the addition of new classes, which gradually fills the latent space and overlaps with existing ones*. This overlap causes confusion in distinguishing between classes, thus reducing performance over time. While existing CL methods emphasize the importance of representation learning to keep classes distinct, none explicitly acknowledge the overlap between new and old tasks as a source of forgetting. Recognizing this motivates us to propose a novel method, focusing on identifying easily confused class pairs, thereby reducing forgetting and improving performance.

## 4 PROPOSED METHOD

In the previous section, we noted that increased overlap in data representations as more tasks arrive is one of the main reasons for greater confusion in predictions, leading to performance degradation. It is crucial to identify easily confused classes/concepts to effectively enhance their distinguishability. Inspired by cognitive science studies (Schön, 1983; Bransford et al., 2000; Mayer, 2005), showing that organizing concepts in a tree-like taxonomy of visually and semantically related items aids memory and retrieval, we propose using expert/domain knowledge to structure the concepts/labels of continual learning (CL) tasks in a hierarchical taxonomy. Interestingly, we find that concepts/labels within the same group in this taxonomy tend to be visually and semantically similar, potentially causing more overlap in the latent space and confusion for the CL classifier. Motivated by this observation, we propose group-based contrastive learning to maximize the distinguishability of these concepts/labels.

### 4.1 MOTIVATION

**Insights from Cognitive Science.** Research in Cognitive Science highlights the importance of *reflection, organization, and linking information* as critical components of effective learning. Studies

show that when learners take time to reflect on their experiences, they deepen their understanding and enhance retention (Schön, 1983). This reflective practice encourages individuals to connect new information with existing knowledge, fostering a more integrated learning experience (Bransford et al., 2000). Moreover, organizing information into coherent structures, such as outlines or concept maps, allows learners to see relationships between concepts, making it easier to retrieve information later (Mayer, 2005). Linking new material with relevant prior knowledge—often referred to as associative learning—further strengthens memory retention (Sweller, 1988) and also benefits future learning.

**Our Approach.** It is evident that besides *reflection*, comparison of old and new information, the key factor in learning efficiently is *to organize and exploit them in an insightful way*, where concepts are linked and arranged according to their semantic meanings. This observation motivates us to develop a deep learning classifier that learns labels or concepts structured in a hierarchical taxonomy. The aim is to enable the classifier to grasp relevant concepts more effectively, helping to mitigate the challenge of catastrophic forgetting. More specifically, we propose structuring the data labels in a hierarchical taxonomy, which can be dynamically constructed using domain expertise, adapting as needed based on the specific context and evolving understanding of the domain. Based on this structure, we establish a reference framework for the relationships between classes, identifying which classes belong to the same group with many shared characteristics, easily confused due to overlap, and require more focus (see Figures 3, 4). This approach not only helps the model better avoid forgetting, but also reinforces knowledge to facilitate future learning.

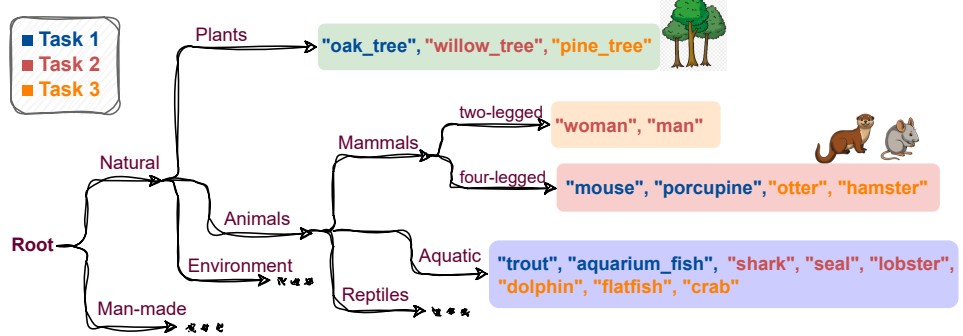

Figure 3: The hierarchical taxonomy obtained when learning Task 3 on Split-CIFAR100.

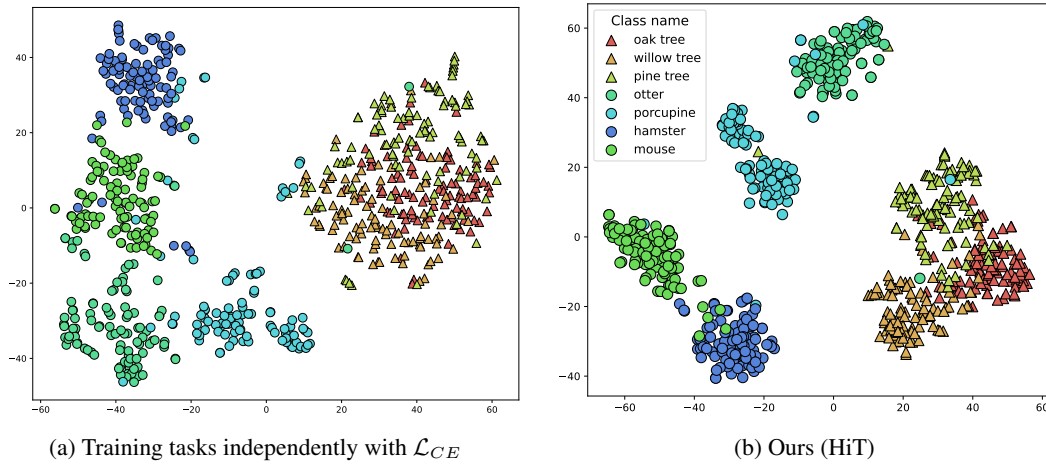

(a) Training tasks independently with $\mathcal{L}_{CE}$      (b) Ours (HiT)

Figure 4: t-SNE visualization of classes within leaf groups of Four-legged animals (● circular points) and Plants (▲ triangular points) when learning Task 3, Split-CIFAR100.

Taking the learning process of Split-CIFAR100 as an example, when training on task $t = 3$, we can construct a tree-like taxonomy of concepts/labels, as shown in Figure 3. We observe that the concepts/classes under the same leaf in the tree-like taxonomy (e.g., oak tree, willow tree, and pine tree in the "plants" leaf group) exhibit stronger visual and semantic correlations. Consequently, as shown in Figure 4a, their features in the feature space become more overlapping compared to those from other leaf groups (e.g., otter and hamster under the "four-legged" leaf group), leading to more confusion and performance degradation when predicting these concepts/classes. This highlights the impact of organizing concepts/labels in a tree-like taxonomy, where data examples within the same leaf group share stronger visual and semantic relationships, causing greater overlap and increased confusion during predictions. Linking to our analysis in Section 3.2, the tree-like taxonomy of concepts/labels serves as a tool to help identify easily confused classes/concepts, facilitating the subsequent process of making them more distinct and separable in the feature space.

Our approach aims to train the backbone network so that all class representations must be distinct, especially those within each leaf group, to achieve maximum distinguishability. For the leaf group of four-legged animals, we assume that "mouse" and "porcupine" arrive in Task 1, while "otter" and "hamster" are in Task 3. When learning Task 1, to classify "mouse" and "porcupine", the backbone is encouraged to capture the essential features of four-legged mammals to efficiently differentiate between these two animals. We hope that the knowledge learned from "mouse" and "otter" in Task 1 can be beneficial for the next tasks. Then in Task 3, we again learn to distinguish "hamster", "otter" and these old ones in this group. In this way, the mechanism helps further strengthen the learning of more efficient and robust features for the Mammals group.

To summarize, by grouping and categorizing, we expect the model to concentrate more on the detailed features of each leaf group. This enhances its ability to distinguish related objects and reinforces the model's knowledge of each group. Therefore, this strategy not only alleviates the forgetting of old knowledge—often caused by new classes that are difficult to distinguish from old ones within the same leaf group, but also enables active knowledge transfer between tasks.

## 4.2 TRAINING PHASE

### 4.2.1 EXPLOITING HIERARCHICAL LABEL/CONCEPT TAXONOMY

During the training process, whenever a new class appears, its label name is automatically added to the tree-like taxonomy, into a leaf group containing classes with similar characteristics (Figure 3). To develop this hierarchical structure, we can rely on expert knowledge, for example, ChatGPT, which can help us incrementally construct a meaningful and semantically-related tree (see Appendix B). Structuring information in this way not only aligns with how the human brain effectively connects and remembers information but also provides useful insights during training, indicating how each knowledge is related to the other and which requires further focus.

As analyzed above, reflecting on and organizing knowledge is the key factor for efficient learning. That is, the model should always be encouraged to identify the decision boundary between all old and new classes, especially those in the same leaf group. Assume that we finished the task $t - 1$ and are learning the prompt $\boldsymbol{P}_t$ for task $t$, our aim is to learn the backbone network $f_{\Phi, \boldsymbol{P}_t}$ that can minimize overlap between all classes so far, especially focus on increasing the separability between classes belonging to the same leaf group extracted from the taxonomy (e.g., four-legged mammals, plants, etc.,). Let $g \in \mathcal{G}$ be a leaf group, $X_k^g$ and $Y_k^g$ denote the corresponding sets of input samples and labels under the group $g$ that belong to the task $k$ $(k \leq t)$. Besides Cross Entropy loss $\mathcal{L}_{CE}$, we propose using a regularization loss function for

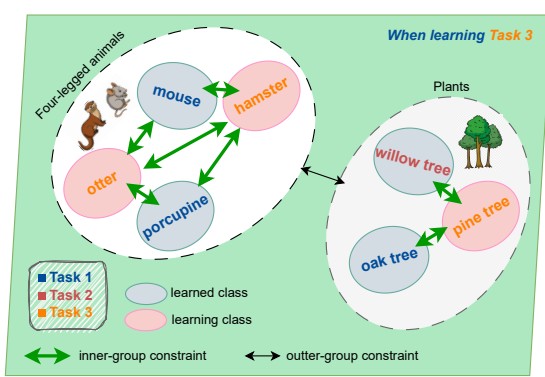

Figure 5: We focus on separate easily confused classes within each leaf group.

sample $\boldsymbol{x}$ (arrives in task $t$, belong leaf group $g$) as follows:

$$\mathcal{L}_{\mathcal{G}}(\psi, \boldsymbol{P}_t, \boldsymbol{x}) = -\alpha \log \sum_{\boldsymbol{x}' \in X_t^g | y_{\boldsymbol{x}'} = y_{\boldsymbol{x}}} \frac{u(\boldsymbol{z}_x \cdot \boldsymbol{z}_{x'})}{\sum_{\bar{x} \in X_{1,t}^g} u(\boldsymbol{z}_x \cdot \boldsymbol{z}_{\bar{x}})} - \beta \mathcal{L}_{all}, \tag{1}$$

where $\mathcal{L}_{all} = \log \sum_{\boldsymbol{x}' \in X_t^g | y_{\boldsymbol{x}'} = y_{\boldsymbol{x}}} \dfrac{u(\boldsymbol{z}_x \cdot \boldsymbol{z}_{x'})}{\sum_{\bar{x} \in X_{\overline{1,t}}} u(\boldsymbol{z}_x \cdot \boldsymbol{z}_{\bar{x}})}$ is the Supervised Contrastive loss that we force

all class representations so far to separate from each other, $u(\boldsymbol{z}_{\boldsymbol{x}} \cdot \boldsymbol{z}_{\boldsymbol{x}'}) = \exp(\frac{\boldsymbol{z}_{\boldsymbol{x}} \cdot \boldsymbol{z}_{\boldsymbol{x}'}}{\tau})$, with $\boldsymbol{z}_{\boldsymbol{x}} = f_{\Phi, \boldsymbol{P}_t}(\boldsymbol{x})$ is the feature vector on the latent space of the prompt-based model, $y_{\boldsymbol{x}}$ is the ground truth label of $\boldsymbol{x}$, $\tau$ is a temperature ($\tau = 0.1$ for all experimental setting), and $\alpha$ is the coefficient that controls how much we want to force on classes belonging to the same leaf group stay apart further. For each data sample $\boldsymbol{x}|y_{\boldsymbol{x}} \in Y_{k<t}$, the corresponding representation $\boldsymbol{z}_{\boldsymbol{x}}$ is sampled from the Gaussian Mixture model $\text{GMM}_{y_{\boldsymbol{x}}} = \{\mathcal{N}(\mu_{y_{\boldsymbol{x}}}, \Sigma_{y_{\boldsymbol{x}}})\}_{k=1}^K$ of the respective class, which is obtained at the end of each corresponding task. This technique of using pseudo features of old data is also employed in many existing prompt-based CL methods, as the shift of features is minimal (Table 1).

Equation 1 implies that when learning a new task, new classes/new knowledge will be compared and contrasted with existing ones. That is, the model will be encouraged to identify the decision boundary between old and new classes, especially focusing on those in the same leaf group via controllable coefficient $\alpha$ (Figure 5). Furthermore, focusing on the specific knowledge within each leaf group helps our model strengthen and consolidate its understanding of this domain, especially when the current prompt is initialized by the previous ones. This is achieved by employing a prompt ensemble strategy similar to that in (Wang et al., 2023): $\boldsymbol{P}_t = \eta \boldsymbol{P}_t' + (1 - \eta) \sum_{i=1}^{t-1} \boldsymbol{P}_i'$, where $\{\boldsymbol{P}_i'\}_{i=1}^t$ is the set of learnable prompt elements which are used to adapt to corresponding tasks, the prompt elements of previous tasks are kept fixed ($\eta = 0.99$ for all setting).

### 4.2.2 AN OPTIMAL TRANSPORT-BASED APPROACH TO FURTHER EXPLOITING PRIORI FROM PRETRAINED MODEL

Considering a leaf group, there may be data classes with varying levels of overlap in the latent space(Figure 4, 6). Although focusing on classes within the same leaf group helps improve the ability to recognize difficult-to-identify classes, we still treat all classes in that group equally. Thus, the algorithm may inadvertently ignore important pairs of classes that are easily confused and need to be further distinguished. Besides, pretrained models are known to have been extensively trained on large datasets, resulting in a substantial repository of generalization ability. Therefore, the prior knowledge from these models often provides a valuable starting point for the adaptation to downstream tasks. However, we seem to frequently overlook the initial behavior of pretrained models on the training data, particularly regarding relations between classes, which classes are easily classified and which are prone to confusion.

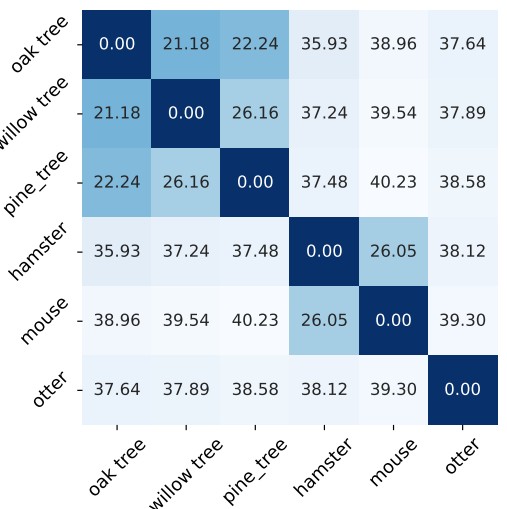

Figure 6: Wasserterin distance between classes (Split-CIFAR100) in latent space of pre-trained backbone (Sup-21K).

Therefore, in this work, we propose to exploit the pre-trained model from a new perspective, which can comprehend the use of the label-based tree-like taxonomy during training, where we can take advantage of prior assumptions about the relationships between the image classes. Firstly, to extract the relationship between the classes, we use L2 Wasserstein distance (WD) to compare the distributions of feature vectors of each pair of class. In particular, let $D_{c_i}^{\Phi}$ be the distribution of class $c_i$ on the latent space of the pretrained model $f_{\Phi}$, which is obtained in the form of a Gaussian Mixture model at the end of the respective task. When $t$ tasks have arrived, we have the corresponding sets

of distributions $\{D_c^\Phi\}_{c \in Y_{1,t}}$ of all $m_t$ classes from all tasks so far. Therefore, we gradually complete the WD-based matrix between pairs of classes:

$$M = [W_2(D_{c_i}^\Phi, D_{c_j}^\Phi)]_{m_t \times m_t}. \tag{2}$$

We then compute *the weight matrix* $\Gamma = [\gamma_{ij}]_{m_t \times m_t} = [1/\exp(M_{ij}/\delta)]_{m_t \times m_t}$, where $\delta$ is a temperature. We then apply this information to obtain a weighted version of $\mathcal{L}_\mathcal{G}$, in which the closer the two class distributions are, the larger the weight assigned, and they will be focused to push away. Consequently, our regularization loss becomes:

$$\mathcal{L}_\mathcal{G}(\psi, \boldsymbol{P}_t, \boldsymbol{x}) = -\alpha \log \sum_{\boldsymbol{x}' \in X_t^g | y_{\boldsymbol{x}'} = y_{\boldsymbol{x}}} \frac{u(\boldsymbol{z}_x \cdot \boldsymbol{z}_{x'})}{\sum_{\bar{x} \in X_{1,t}^g} \gamma_{y_x y_{\bar{x}}} u(\boldsymbol{z}_x \cdot \boldsymbol{z}_{\bar{x}})} - \beta \mathcal{L}_{all}. \tag{3}$$

This strategy is completely economical and aligns well with the CL learning scheme as matric $M$ is continuously expanded and provides useful information for training new tasks. Practically, when learning a new task, the first epoch is spent capturing information about the behavior of the pretrained model on the data for this task. Moreover, this approach is similar to the findings in Cognitive Science (Osgood & Bower, 1953; Baltes, 1987), showing that the accumulated experiences from past learning create momentum for learning new skills more effectively.

### 4.3 TESTING PHASE

We observe that classes within the same leaf group often share many common characteristics. Therefore, we propose a testing strategy that leverages information from each leaf group to gain a new perspective on the identification of data samples, especially those at the boundaries between different leaf groups.

In particular, the final prediction for sample $\boldsymbol{x}$ will be modified based on the probability that $\boldsymbol{x}$ belongs to a certain leaf group $g \in \mathcal{G}$ (i.e., $p(g|\boldsymbol{x}, \boldsymbol{P}_{\hat{t}})$). Intuitively, if the representation $\boldsymbol{z}_x$ of $\boldsymbol{x}$ has many similarities with class representations in group $g$, then $p(g|\boldsymbol{x}, \boldsymbol{P}_{\hat{t}})$ will increase, thereby raising the likelihood that $\boldsymbol{x}$ belongs to the corresponding classes in $g$ (i.e., $p(y|g, \boldsymbol{x}, \boldsymbol{P}_{\hat{t}})$):

$$p(y|\boldsymbol{x}) = p(y|\boldsymbol{x}, \boldsymbol{P}_{\hat{t}}) = \sum_{g \in \mathcal{G}} p(g|\boldsymbol{x}, \boldsymbol{P}_{\hat{t}}) \cdot p(y|g, \boldsymbol{x}, \boldsymbol{P}_{\hat{t}}) = \sum_{g \in \mathcal{G}} p(g|\boldsymbol{x}, \boldsymbol{P}_{\hat{t}}) \cdot p(y|x, \boldsymbol{P}_{\hat{t}}) \cdot \mathbb{I}_{y \in Y^g}, \tag{4}$$

where $\mathbb{I}_{y \in Y^g} = 1$ if $y \in Y^g$, else 0. The value of $p(g|\boldsymbol{x}, \boldsymbol{P}_{\hat{t}})$ is calculated based on the "energy" of $\boldsymbol{x}$ w.r.t group $g$, in relation to other group $g' \neq g$:

$$p(g|\boldsymbol{x}, \boldsymbol{P}_{\hat{t}}) = \frac{\exp\{E(\boldsymbol{x}, g)\}}{\sum_{g' \in \mathcal{G}} \exp\{E(\boldsymbol{x}, g')\}}. \tag{5}$$

In Eq. (5), $E(\boldsymbol{x}, g)$ indicates the "energy" of $\boldsymbol{x}$ w.r.t leaf group $g \in \mathcal{G}$. Remind that for each class $c$, we maintain a GMM of $K$ mixtures $\{\mathcal{N}(\boldsymbol{\mu}_{c,i}, \Sigma_{c,i})\}_{i=1}^K$. Based on the prototypes for a class, we can define the distance from $\boldsymbol{x}$ to a class $c$ as $d(\boldsymbol{x}, c) = \min_{1 \leq i \leq K} cosine\_distance(\boldsymbol{z}_x, \boldsymbol{\mu}_{c,i})$. Limiting to the group $g$, we define $\hat{y}_{\boldsymbol{x}}^g = \arg\min_{c \in Y^g} d(\boldsymbol{x}, c)$ (i.e., $Y^g$ is the set of all classes in $g$). We define the energy of interest as

$$E(\boldsymbol{x}, g) = -d(\boldsymbol{x}, \hat{y}_{\boldsymbol{x}}^g) - \xi \sum_{c \in Y^g} \gamma_{c, \hat{y}_{\boldsymbol{x}}^g} \sum_{i=1}^K \sqrt{(\boldsymbol{z}_{\boldsymbol{x}} - \mu_{c,i})^T \Sigma_{c,i}^{-1} (\boldsymbol{z}_{\boldsymbol{x}} - \mu_{c,i})} \tag{6}$$

where the first terms is the cosine similarity between $\boldsymbol{z}_{\boldsymbol{x}}$ and the closest class prototype within group $g$ and $\gamma_{c, \hat{y}_{\boldsymbol{x}}^g}$ is the value obtained from *the weight matrix* $\Gamma$ (Section 4.2.2) - indicating the correlation between class $c$. This approach exploits the correlation between $\boldsymbol{x}$ and $g$ while reducing the disadvantage of large groups with many classes, whereby the distances of classes that are less related to $y$ will have less weight and vice versa. Finally, $\xi$ is the hyperparameter, which controls the amount of information referenced from the group.

By this strategy, features $\boldsymbol{z}_{\boldsymbol{x}}$ will have an additional point of view to determine which class that $\boldsymbol{x}$ is more likely to belong to, especially for $\boldsymbol{z}_{\boldsymbol{x}}$ located at the boundary between leaf groups—such as between the group of "Plants" and the group of "Four-legged animals". This is similar to how having more prior knowledge improves posterior probability in Bayes' rule and how humans with more in-depth knowledge in different fields have greater experience in solving problems.

## 5 EXPERIMENTS

### 5.1 EXPERIMENTAL SETUP

**Datasets.** We examine widely used CIL benchmarks, including Split CIFAR-100, Split ImageNet-R, 5-Datasets, and Split CUB-200 (please refer Appendix A.1 for more details).

**Baselines.** We compare our method with notable CL methods exploiting prompt-based approach for pre-trained models, including the methods using shared prompts for all tasks: L2P (Wang et al., 2022d), DualPrompt (Wang et al., 2022c), OVOR (Huang et al., 2024); and the methods dedicates a distinct prompt set for each task like: S-Prompt++ (Wang et al., 2022b), CODA-Prompt (Smith et al., 2023), HiDe-Prompt (Wang et al., 2023), CPP (Li et al., 2024).

**Metrics.** We use two main metrics, including the Final Average Accuracy (FAA), denoting the average accuracy after learning the last task, and the Final Forgetting Measure (FFM) showing the forgetting of all tasks after learning the sequence of tasks (see Appendices A.2 & A.3).

The implementation is described in detail in Appendix A.4.

### 5.2 EXPERIMENTAL RESULT

**Our approach achieves superior results compared to baselines.** Table 2 presents the overall performance comparison between our proposed method and all the baselines. The key observation is that our method is the strongest one with the gap between our method and the runner-up method is about 2% in terms of FAA on all considered datasets. Additionally, the results show that our method avoids forgetting better than all baselines, notably reducing forgetting by more than 2% on the Split-CIFAR100 dataset compared to the strongest one.

Table 2: Overall performance comparison. We provide FAA and FFM of all methods, with standard deviation taken over at least 3 runs of different random seeds. The results corresponding to the best FAA among baselines are underlined.

| Method | Split CIFAR-100 | | Split ImageNet-R | | 5-Datasets | | Split CUB-200 | |
|---|---|---|---|---|---|---|---|---|
| | **FAA** (↑) | FFM (↓) | **FAA** (↑) | FFM (↓) | **FAA** (↑) | FFM (↓) | **FAA** (↑) | FFM (↓) |
| L2P | 83.06 ±0.17 | 6.58 ±0.40 | 63.65 ±0.12 | 7.51 ±0.17 | 81.84 ±0.95 | 4.58 ±0.53 | 74.52 ±0.92 | 11.25 ±0.23 |
| DualPrompt | 86.60 ±0.19 | 4.45 ±0.16 | 68.79 ±0.31 | 4.49 ±0.14 | 77.91±0.45 | 13.17 ±0.71 | 82.05±0.95 | 3.56 ±0.53 |
| OVOR | 86.68 ±0.22 | 5.25 ±0.12 | 75.61 ±0.82 | 5.77 ±0.12 | 82.34 ±0.48 | 4.83 ±0.35 | 78.12 ±0.0.65 | 8.13 ±0.52 |
| S-Prompt++ | 88.81 ±0.18 | 3.87 ±0.05 | 69.68 ±0.12 | 3.29 ±0.05 | 86.19±0.65 | 4.67 ±0.72 | 83.12 ±0.54 | 2.72 ±0.64 |
| CODA-P | 86.94 ±0.63 | 4.04 ±0.18 | 70.03 ±0.47 | 5.17 ±0.22 | 64.20 ±0.53 | 17.22 ±0.55 | 74.34 ±0.68 | 12.05 ±0.41 |
| CPP | 91.12 ±0.12 | 3.33 ±0.18 | 74.88 ±0.07 | 4.08 ±0.03 | 92.92 ±0.17 | 0.23 ±0.07 | 82.35 ±0.23 | 3.24 ±0.32 |
| HiDe-Prompt | 92.61 ±0.28 | 1.52 ±0.10 | 75.06 ±0.12 | 4.05 ±0.19 | 93.92 ±0.33 | 0.31 ±0.12 | 86.62 ±0.35 | 2.55 ±0.15 |
| Ours (HCL) | **94.52** ±0.22 | **1.02** ±0.18 | **77.01** ±0.12 | **4.03** ±0.25 | **95.35** ±0.18 | **0.20** ±0.16 | **88.33** ±0.18 | **1.98** ±0.22 |

**Our training strategy improves model performance significantly.** Figure 7 reports the ablation studies demonstrating the effectiveness of our training strategy. Particularly, compared to training tasks independently using Cross Entropy loss $\mathcal{L}_{CE}$ like in DualP, L2P, and CODA-P, exploiting the relationships between data classes with the label-based hierarchical taxonomy and the WD-based cost matrix helps improve FAA by about 5% to 10% (Figure 7a). Besides, when examining the role of exploiting additional prior information from pretrained backbones using the OT approach, we see that FAA is improved from 0.6% to 0.8% (Figure 7b). These results demonstrate the positive impact of this component, confirming the importance of exploiting correlations between class data during training. In both figures, the improvements on Split-CIFAR100 and 5-Datasets are the lowest, while it is more pronounced on Split-CUB-200. This may be because the groups of these two datasets (Split-CIFAR100 and 5-Datasets) have fewer overlapping classes, as the classes in each group likely have more recognizable features. Meanwhile, Split-CUB-200 is a dataset about birds, with images that can be difficult for human eyes to recognize, thus so our method performs better.

In addition, Figure 7c provides the experimental results on Split-CUB-200 dataset, when varying $\alpha$ and $\beta$, which control the intensification of impact on each leaf group of $\mathcal{L}_\mathcal{G}$ during training. The data shows that with a large enough value of $\beta$, our HiT is not sensitive to $\alpha$ within its acceptable range. Conversely, if $\alpha$ is small, the quality of the model can change more significantly.

Furthermore, Figure 4 illustrates the effect of our method in improving model's representation learning on Split-CIFAR100. Specifically, the classes are better clustered, and the separation between them is more distinct. Especially, the classes 'oak tree,' 'willow tree,' and 'pine tree' are divided into clear clusters, rather than being mixed together as in the traditional training strategy, where tasks are trained independently.

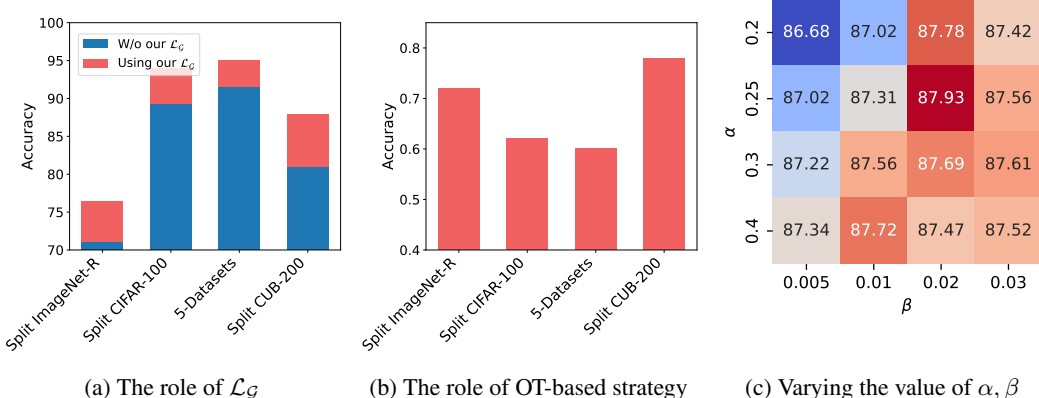

| (a) The role of $\mathcal{L}_\mathcal{G}$ | (b) The role of OT-based strategy | (c) Varying the value of $\alpha, \beta$ |

Figure 7: Ablation study about our training strategy.

**Our testing strategy has positive effects for final prediction**  Table 3 illustrates the improvement of FAA on all considered datasets when applying our testing strategy, from about 0.4% to 0.6%. This proves the approach to be effective, as the information from each cluster provides a reference channel that helps determine the identity of the classes, offering a good suggestion for future studies.

Table 3: Effectiveness of our testing strategy.

| Dataset | Split CIFAR-100 | Split Imagenet-R | 5-Datasets | Split CUB-200 |
|---------|-----------------|------------------|------------|---------------|
| Normal testing | 93.94 | 76.41 | 95.02 | 87.93 |
| Our testing strategy | **94.52** | **77.01** | **95.35** | **88.33** |

# 6  CONCLUSION AND LIMITATION DISCUSSION

In this work, we demonstrate the importance of organizing and exploiting data meaningfully rather than lumping it together for training. Organizing data into a tree-like taxonomy based on label information gives us a new perspective on the data. Particularly, we can divide them into small groups containing the classes that are likely to confuse models. This approach encourages the model to focus and build deeper knowledge for each group, thereby reducing forgetting and motivating more effective learning in subsequent tasks. Additionally, we introduce a new perspective by leveraging the initial behavior of pretrained models, providing an additional information channel to further improve performance. Besides, our testing strategy has shown positive effects by exploiting group knowledge during inference. Finally, experimental results demonstrate the effectiveness of these components and our superiority over state-of-the-art baselines.

Despite this novel perspective, the quality of the hierarchical taxonomy depends on the quality of expert knowledge. For example, if similar image classes are not assigned to the same leaf group in this label-based taxonomy, the constraint we put on each such group may not perform as expected. Furthermore, although the testing strategy shows positive results, to exploit group knowledge more efficiently, it is necessary to further investigate to understand the characteristics and hidden structure of data.

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

# Supplement to "Exploiting prior knowledge for pre-trained CL"

## A  EXPERIMENTAL SETTINGS

### A.1  DATASETS

We adopt the following common benchmarks:

- **Split CIFAR-100** (Krizhevsky et al., 2009): This dataset includes images from 100 different classes, each being relatively small in size. The classes are randomly organized into 10 sequential tasks, with each task containing a unique set of classes.
- **Split ImageNet-R** (Krizhevsky et al., 2009): This dataset contains images from 200 extensive classes. It includes difficult examples from the original **ImageNet** dataset, as well as newly acquired images that display a variety of styles. The classes are randomly divided into 10 distinct incremental tasks.
- **5-Datasets** (Ebrahimi et al., 2020): This composite dataset incorporates **CIFAR-10** (Krizhevsky et al., 2009), **MNIST** (LeCun et al., 1998), **Fashion-MNIST** (Xiao et al., 2017), **SVHN** (Netzer et al., 2011), and **notMNIST** (Bulatov, 2011). Each of these is treated as a separate incremental task, enabling the evaluation of the impact of substantial variations between tasks.
- **Split CUB-200** (Wah et al., 2011): This dataset contains fine-grained images of 200 distinct bird species. It is randomly divided into 10 incremental tasks, each with a unique subset of classes.

### A.2  BASELINES

In the main paper, we use CL methods with pre-trained ViT as the backbone. We group them into (a) the group using a common prompt pool for all tasks, and (b) the group dedicating distinct prompt sets for each task:

(1) **L2P** (Wang et al., 2022d): The first prompt-based work for continual learning (CL) suggested using a common prompt pool, selecting the top $k$ most suitable prompts for each sample during training and testing. This approach might facilitate knowledge transfer between tasks but also risks catastrophic forgetting. Unlike our approach, L2P doesn't focus on training classifiers or setting constraints on features from old and new tasks during training, which may limit the model's predictability.

(2) **DualPrompt** (Wang et al., 2022c): The prompt-based method aims to address L2P's limitations by attaching complementary prompts to the pre-trained backbone, rather than only at input. DualP introduces additional prompt sets for each task to leverage task-specific instructions alongside invariant information from the common pool. However, like L2P, it does not focus on efficiently learning the classification head. Additionally, selecting the wrong prompt ID for task-specific instructions during testing can negatively impact model performance.

(3) **OVOR** (Huang et al., 2024): while using only a common prompt pool for all tasks, this work introduces a regularization method for Class-incremental learning that uses virtual outliers to tighten decision boundaries, reducing confusion between classes from different tasks. Experimental results demonstrate the role of representation learning, which focuses on reducing overlapping between class representations.

(4) **S-Prompt++** (Wang et al., 2022b): S-Prompt was originally proposed for domain-incremental learning, training a separate prompt and classifier head for each task. During evaluation, it infers the domain ID using the nearest centroid from K-Means applied to the training data. To adapt S-Prompt to class-incremental learning (CIL), S-Prompt++ uses a common classifier head for all tasks. However, it shares limitations with DualP, such as efficient learning of the classification head and predicting appropriate prompts during testing.

(5) **CODA-Prompt** (Smith et al., 2023): This prompt-based approach uses task-specific learnable prompts for each task. Similar to L2P, CODA employs a pool of prompts and keys, computing a weighted sum from these prompts to generate the real prompt. The weights are based on the cosine

similarity between queries and keys. To avoid task prediction at the end of the task sequence, the weighted sum always considers all prompts. CODA improves over DualP and L2P by optimizing keys and prompts simultaneously, but it still hasn't addressed the drawbacks mentioned for DualP.

(6) **HiDe-Prompt** (Wang et al., 2023): a recent SOTA prompt-based method that decomposes learning CIL into 3 modules: a task inference, a within-task predictor and a task-adaptive predictor. The second module trains prompts for each task with a contrastive regularization that tries to push features of new tasks away from prototypes of old ones. To predict task identity, it trains a classification head on top of the pre-trained ViT. TAP is similar to a fine-tuning step that aims to alleviate classifier bias using the Gaussian distribution of all classes seen so far. However, this method does not declare the relationship between data during training, thereby missing the opportunity to improve model performance.

(7) **CPP** (Li et al., 2024): This recent SOTA also uses a contrastive constraint to control features of all tasks so far during representation learning and achieves roughly equivalent performance to HiDE on the same settings. Nevertheless, this method still has the advantages that we pointed out in HiDE, which we propose to address in our work.

## A.3 METRICS

In our study, we employed two key metrics: the Final Average Accuracy (FAA) and the Final Forgetting Measure (FFM). To define these, we first consider the accuracy on the $i$-th task after the model has been trained up to the $t$-th task, denoted as $A_{i,t}$. The average accuracy of all tasks observed up to the $t$-th task is calculated as $AA_t = \frac{1}{t} \sum_{i=1}^{t} A_{i,t}$. Upon the completion of all $T$ tasks, we report the Final Average Accuracy as FAA $= AA_T$. Additionally, we calculate the Final Forgetting Measure, defined as FFM $= \frac{1}{T-1} \sum_{i=1}^{T-1} \max_{t \in \{1,...,T-1\}} (A_{i,t} - A_{i,T})$. The FAA serves as the principal indicator for assessing the ultimate performance in continual learning models, while the FFM evaluates the extent of catastrophic forgetting experienced by the model.

## A.4 IMPLEMENTATION DETAILS

Our implementation basically aligns with the methodologies employed in prior research Wang et al. (2022d;c); Smith et al. (2023). Specifically, our framework incorporates the use of a pre-trained Vision Transformer (ViT-B/16) as the backbone architecture. For the optimization process, we utilized the Adam optimizer, configured with hyper-parameters $\beta_1$ set to $0.9$ and $\beta_2$ set to $0.999$. The training process was conducted using batches of $24$ samples, and a fixed learning rate of $0.03$ was applied across all models except for CODA-Prompt. For CODA-Prompt, we employed a cosine decaying learning rate strategy, starting at $0.001$. Additionally, a grid search technique was implemented to determine the most appropriate number of epochs for effective training. Regarding the pre-processing of input data, images were resized to a standard dimension of $224 \times 224$ pixels and normalized within a range of $[0, 1]$ to ensure consistency in input data format. The detailed values of the parameters can be found in our source code.

In Table 2 of the main paper, the results of L2P, DualPrompt, S-Prompt++, CODA-Prompt, and HiDe-Prompt on Split CIFAR-100 and Split ImageNet-R are taken from (Wang et al., 2023). Their results on the other two datasets are produced from the official code provided by the authors. For CPP and OVOR, the reported results are also reproduced from their official code. It's worth noting that the reported forgetting of HiDE is reproduced from their official code.

## B USING CHATGPT TO BUILD TREE-LIKE TAXONOMY DURING A SEQUENCE OF TASKS INCREMENTALLY

We use the following prompt structure to generate the taxonomies:

```
Given the label list: ['···'], provide me the taxonomy from this
list, based on their origin, type, and shape, so that the image
encoders can recognize their images.
```

Example output, when the list ['···'] is ["leopard", "rabbit", "mouse", "camel", "trout", "aquarium_fish", "snake", "rose", "lawn_mower", "bottle"]:

```
taxonomy = {
    "Natural": {
        "Animals": {
            "Mammals": {
                "Four-legged": ["leopard", "rabbit", "mouse", "camel"]
            },
            "Aquatic": ["trout", "aquarium_fish"],
            "Reptiles": ["snake"]
        },
        "Plants": {
            "Flowers": ["rose"]
        }
    },
    "Man-Made": {
        "Objects": {
            "Tools": ["lawn_mower"],
            "Containers": ["bottle"]
        }
    }
}
```

Below is an example of generated taxonomies for each task of Split-CIFAR100:

```
T1 = {
    "Natural": {
        "Animals": {
            "Mammals": {
                "Four-legged": ["leopard", "rabbit", "mouse", "camel"]
            },
            "Aquatic": ["trout", "aquarium_fish"],
            "Reptiles": ["snake"]
        },
        "Plants": {
            "Flowers": ["rose"]
        }
    },
    "Man-Made": {
        "Objects": {
            "Tools": ["lawn_mower"],
            "Containers": ["bottle"]
        }
    }
}

T2 = {
    "Natural": {
        "Animals": {
            "Mammals": {
                "Four-legged": ["leopard", "rabbit",
                                "mouse", "camel", "otter"]
            },
            "Aquatic": ["trout", "aquarium_fish",
                        "shark", "seal", "lobster"],
            "Reptiles": ["snake"]
        },
        "Plants": {
            "Flowers": ["rose", "tulip"],
            "Trees": ["palm_tree"]
```

```
           }
       },
       "Man-Made": {
           "Objects": {
               "Tools": ["lawn_mower"],
               "Containers": ["bottle", "bowl"]
           },
           "Vehicles": {
               "Wheeled": ["motorcycle"]
           },
           "Structures": {
               "Buildings": ["skyscraper", "house"]
           }
       }
   }

   T3 = {
       "Natural": {
           "Animals": {
               "Mammals": {
                   "Four-legged": ["leopard", "rabbit", "mouse",
                               "camel", "otter", "chimpanzee", "squirrel"]
               },
               "Aquatic": ["trout", "aquarium_fish", "shark",
                       "seal", "lobster", "dolphin", "flatfish", "crab"],
               "Reptiles": ["snake"]
           },
           "Plants": {
               "Flowers": ["rose", "tulip"],
               "Trees": ["palm_tree", "willow_tree"],
               "Fruits": ["sweet_pepper"]
           },
           "Environment": {
               "Natural Features": ["mountain", "forest"]
           }
       },
       "Man-Made": {
           "Objects": {
               "Tools": ["lawn_mower"],
               "Containers": ["bottle", "bowl"],
               "Appliances": ["television"]
           },
           "Vehicles": {
               "Wheeled": ["motorcycle"]
           },
           "Structures": {
               "Buildings": ["skyscraper", "house"]
           }
       }
   }

   T4 = {
       "Natural": {
           "Animals": {
               "Mammals": {
                   "Four-legged": [
                       "leopard", "rabbit", "mouse", "camel", "otter",
                       "chimpanzee", "squirrel", "porcupine", "shrew"
                       ],
```

```
                    "Two-legged": ["woman"]
                },
                "Aquatic": [
                    "trout", "aquarium_fish", "shark", "seal", "lobster",
                    "dolphin", "flatfish", "crab"
                ],
                "Reptiles": ["snake", "lizard"]
            },
            "Plants": {
                "Flowers": ["rose", "tulip"],
                "Trees": ["palm_tree", "willow_tree",
                        "maple_tree", "pine_tree", "oak_tree"],
                "Fruits": ["sweet_pepper"]
            },
            "Environment": {
                "Natural Features": ["mountain", "forest"]
            }
        },
        "Man-Made": {
            "Objects": {
                "Tools": ["lawn_mower", "tank"],
                "Containers": ["bottle", "bowl"],
                "Appliances": ["television"]
            },
            "Vehicles": {
                "Wheeled": ["motorcycle", "bicycle"]
            },
            "Structures": {
                "Buildings": ["skyscraper", "house"],
                "Bridges": ["bridge"]
            }
        }
    }

T5 = {
    "Natural": {
        "Animals": {
            "Mammals": {
                "Four-legged": [
                    "leopard", "rabbit", "mouse", "camel", "otter",
                    "chimpanzee", "squirrel", "porcupine", "shrew",
                    "hamster", "raccoon", "fox"
                ],
                "Two-legged": ["woman"]
            },
            "Aquatic": [
                "trout", "aquarium_fish", "shark", "seal", "lobster",
                "dolphin", "flatfish", "crab"
            ],
            "Reptiles": ["snake", "lizard"],
            "Insects": ["caterpillar", "beetle"]
        },
        "Plants": {
            "Flowers": ["rose", "tulip"],
            "Trees": ["palm_tree", "willow_tree",
                    "maple_tree", "pine_tree", "oak_tree"],
            "Fruits": ["sweet_pepper"]
        },
        "Environment": {
```

```
972                  "Natural Features": ["mountain", "forest", "cloud", "plain"]
973             }
974         },
975         "Man-Made": {
976             "Objects": {
977                 "Tools": ["lawn_mower", "tank"],
978                 "Containers": ["bottle", "bowl", "plate"],
979                 "Appliances": ["television"],
980                 "Instruments": ["keyboard", "lamp"]
981             },
982             "Vehicles": {
983                 "Wheeled": ["motorcycle", "bicycle"]
984             },
985             "Structures": {
986                 "Buildings": ["skyscraper", "house"],
987                 "Bridges": ["bridge"]
988             }
989         }
990     }
991
992     T6 = {
993         "Natural": {
994             "Animals": {
995                 "Mammals": {
996                     "Four-legged": [
997                         "leopard", "rabbit", "mouse", "camel", "otter",
998                         "chimpanzee", "squirrel", "porcupine", "shrew",
999                         "hamster", "raccoon", "fox", "kangaroo"
1000                    ],
1001                    "Two-legged": ["woman", "man", "baby"]
1002                },
1003                "Aquatic": [
1004                    "trout", "aquarium_fish", "shark", "seal", "lobster",
1005                    "dolphin", "flatfish", "crab"
1006                ],
1007                "Reptiles": ["snake", "lizard"],
1008                "Insects": ["caterpillar", "beetle"],
1009                "Others": ["worm"]
1010            },
1011            "Plants": {
1012                "Flowers": ["rose", "tulip", "poppy"],
1013                "Trees": ["palm_tree", "willow_tree",
1014                        "maple_tree", "pine_tree", "oak_tree"],
1015                "Fruits": ["sweet_pepper"],
1016                "Fungi": ["mushroom"]
1017            },
1018            "Environment": {
1019                "Natural Features": ["mountain", "forest", "cloud", "plain"]
1020            }
1021        },
1022        "Man-Made": {
1023            "Objects": {
1024                "Tools": ["lawn_mower", "tank"],
1025                "Containers": ["bottle", "bowl", "plate", "can"],
                   "Appliances": ["television"],
                   "Instruments": ["keyboard", "lamp", "clock"]
               },
               "Vehicles": {
                   "Wheeled": ["motorcycle", "bicycle", "pickup_truck"]
```

```
1026            },
1027            "Structures": {
1028                "Buildings": ["skyscraper", "house"],
1029                "Bridges": ["bridge"],
1030                "Others": ["road"]
1031            }
1032        }
1033    }
1034
1035    T7 = {
1036        "Natural": {
1037            "Animals": {
1038                "Mammals": {
1039                    "Four-legged": [
1040                        "leopard", "rabbit", "mouse", "camel", "otter",
1041                        "chimpanzee", "squirrel", "porcupine", "shrew",
1042                        "hamster", "raccoon", "fox", "kangaroo", "cattle", "lion"
1043                    ],
1044                    "Two-legged": ["woman", "man", "baby"]
1045                },
1046                "Aquatic": [
1047                    "trout", "aquarium_fish", "shark", "seal",
1048                    "lobster", "dolphin", "flatfish", "crab", "ray"
1049                ],
1050                "Reptiles": ["snake", "lizard"],
1051                "Insects": ["caterpillar", "beetle", "bee", "cockroach", "spider"],
1052                "Others": ["worm"]
1053            },
1054            "Plants": {
1055                "Flowers": ["rose", "tulip", "poppy", "sunflower"],
1056                "Trees": ["palm_tree", "willow_tree",
1057                        "maple_tree", "pine_tree", "oak_tree"],
1058                "Fruits": ["sweet_pepper"],
1059                "Fungi": ["mushroom"]
1060            },
1061            "Environment": {
1062                "Natural Features": ["mountain", "forest", "cloud", "plain"]
1063            }
1064        },
1065        "Man-Made": {
1066            "Objects": {
1067                "Tools": ["lawn_mower", "tank"],
1068                "Containers": ["bottle", "bowl", "plate", "can"],
1069                "Appliances": ["television"],
1070                "Instruments": ["keyboard", "lamp", "clock"],
1071                "Furniture": ["bed", "chair"]
1072            },
1073            "Vehicles": {
1074                "Wheeled": ["motorcycle", "bicycle", "pickup_truck"],
1075                "Rail": ["train"]
1076            },
1077            "Structures": {
1078                "Buildings": ["skyscraper", "house"],
1079                "Bridges": ["bridge"],
1080                "Others": ["road"]
1081            }
1082        }
1083    }
```

```
T8 = {
    "Natural": {
        "Animals": {
            "Mammals": {
                "Four-legged": [
                    "leopard", "rabbit", "mouse", "camel", "otter",
                    "chimpanzee", "squirrel", "porcupine", "shrew",
                    "hamster", "raccoon", "fox", "kangaroo", "cattle", "lion"
                ],
                "Two-legged": ["woman", "man", "baby"]
            },
            "Aquatic": [
                "trout", "aquarium_fish", "shark", "seal", "lobster",
                "dolphin", "flatfish", "crab", "ray", "whale"
            ],
            "Reptiles": ["snake", "lizard", "turtle"],
            "Insects": ["caterpillar", "beetle", "bee", "cockroach", "spider"],
            "Others": ["worm", "snail"]
        },
        "Plants": {
            "Flowers": ["rose", "tulip", "poppy", "sunflower"],
            "Trees": ["palm_tree", "willow_tree",
                        "maple_tree", "pine_tree", "oak_tree"],
            "Fruits": ["sweet_pepper", "apple", "pear", "orange"],
            "Fungi": ["mushroom"]
        },
        "Environment": {
            "Natural Features": ["mountain", "forest",
                        "cloud", "plain", "sea"]
        }
    },
    "Man-Made": {
        "Objects": {
            "Tools": ["lawn_mower", "tank"],
            "Containers": ["bottle", "bowl", "plate", "can"],
            "Appliances": ["television"],
            "Instruments": ["keyboard", "lamp", "clock"],
            "Furniture": ["bed", "chair", "couch", "table"]
        },
        "Vehicles": {
            "Wheeled": ["motorcycle", "bicycle", "pickup_truck", "tractor"],
            "Rail": ["train"]
        },
        "Structures": {
            "Buildings": ["skyscraper", "house"],
            "Bridges": ["bridge"],
            "Others": ["road"]
        }
    }
}

T9 = {
    "Natural": {
        "Animals": {
            "Mammals": {
                "Four-legged": [
                    "leopard", "rabbit", "mouse", "camel", "otter",
                    "chimpanzee", "squirrel", "porcupine", "shrew",
                    "hamster", "raccoon", "fox", "kangaroo", "cattle",
```

```
                            "lion", "tiger", "wolf", "beaver", "possum", "skunk"
                        ],
                        "Two-legged": ["woman", "man", "baby", "boy"]
                    },
                    "Aquatic": [
                        "trout", "aquarium_fish", "shark", "seal", "lobster",
                        "dolphin", "flatfish", "crab", "ray", "whale"
                    ],
                    "Reptiles": ["snake", "lizard",
                                 "turtle", "crocodile", "dinosaur"],
                    "Insects": ["caterpillar", "beetle",
                                 "bee", "cockroach", "spider"],
                    "Others": ["worm", "snail"]
                },
                "Plants": {
                    "Flowers": ["rose", "tulip", "poppy", "sunflower", "orchid"],
                    "Trees": ["palm_tree", "willow_tree",
                                 "maple_tree", "pine_tree", "oak_tree"],
                    "Fruits": ["sweet_pepper", "apple", "pear", "orange"],
                    "Fungi": ["mushroom"]
                },
                "Environment": {
                    "Natural Features": ["mountain", "forest",
                                         "cloud", "plain", "sea"]
                }
            },
            "Man-Made": {
                "Objects": {
                    "Tools": ["lawn_mower", "tank"],
                    "Containers": ["bottle", "bowl", "plate", "can"],
                    "Appliances": ["television"],
                    "Instruments": ["keyboard", "lamp", "clock"],
                    "Furniture": ["bed", "chair", "couch", "table"]
                },
                "Vehicles": {
                    "Wheeled": ["motorcycle", "bicycle",
                             "pickup_truck", "tractor"],
                    "Air": ["rocket"],
                    "Rail": ["train"]
                },
                "Structures": {
                    "Buildings": ["skyscraper", "house"],
                    "Bridges": ["bridge"],
                    "Others": ["road"]
                }
            }
        }

        T10 = {
            "Natural": {
                "Animals": {
                    "Mammals": {
                        "Four-legged": [
                            "leopard", "rabbit", "mouse", "camel", "otter",
                            "chimpanzee", "squirrel", "porcupine", "shrew",
                            "hamster", "raccoon", "fox", "kangaroo", "cattle",
                            "lion", "tiger", "wolf", "beaver", "possum", "skunk",
                            "elephant", "bear"
                        ],
```

```
                    "Two-legged": ["woman", "man",
                           "baby", "boy", "girl"]
                },
                "Aquatic": [
                    "trout", "aquarium_fish", "shark", "seal", "lobster",
                    "dolphin", "flatfish", "crab", "ray", "whale"
                ],
                "Reptiles": ["snake", "lizard",
                        "turtle", "crocodile", "dinosaur"],
                "Insects": ["caterpillar", "beetle",
                            "bee", "cockroach", "spider", "butterfly"],
                "Others": ["worm", "snail"]
            },
            "Plants": {
                "Flowers": ["rose", "tulip", "poppy", "sunflower", "orchid"],
                "Trees": ["palm_tree", "willow_tree",
                            "maple_tree", "pine_tree", "oak_tree"],
                "Fruits": ["sweet_pepper", "apple", "pear", "orange"],
                "Fungi": ["mushroom"]
            },
            "Environment": {
                "Natural Features": ["mountain", "forest",
                        "cloud", "plain", "sea"]
            }
        },
        "Man-Made": {
            "Objects": {
                "Tools": ["lawn_mower", "tank"],
                "Containers": ["bottle", "bowl", "plate", "can", "cup"],
                "Appliances": ["television"],
                "Instruments": ["keyboard", "lamp", "clock", "telephone"],
                "Furniture": ["bed", "chair", "couch", "table", "wardrobe"]
            },
            "Vehicles": {
                "Wheeled": ["motorcycle", "bicycle",
                        "pickup_truck", "tractor", "bus"],
                "Rail": ["train", "streetcar"],
                "Air": ["rocket"]
            },
            "Structures": {
                "Buildings": ["skyscraper", "house", "castle"],
                "Bridges": ["bridge"],
                "Others": ["road"]
            }
        }
    }
}
```

The taxonomies for other datasets are available in our source code.

## C    ADDITIONAL EXPERIMENTS

**The superiority of our proposed method on various types of pre-trained backbones.**    Table 4 illustrates that our method with the training strategy only consistently outperforms the strongest baseline (HiDE) by the gap from about 0.5% to 1.5% in all cases.

Table 4: Comparison when using different pre-trained backbones

| Pretrained backbone | Split CIFAR-100 | | Split Imagenet-R | |
|---|---|---|---|---|
| | HiT | HiDE | HiT | HiDE |
| Sup-21K | **93.94** (↑ 1.33) | 92.61 | **76.41** (↑ 1.35) | 75.06 |
| iBOT-21K | **94.01** (↑ 0.99) | 93.02 | **72.12** (↑ 1.29) | 70.83 |
| iBOT-1K | **94.27** (↑ 0.79) | 93.48 | **72.80** (↑ 1.47) | 71.33 |
| DINO-1K | **94.12** (↑ 0.61) | 93.51 | **69.25** (↑ 1.14) | 68.11 |
| MoCo-1K | **92.32** (↑ 0.75) | 91.57 | **64.23** (↑ 0.46) | 63.77 |

