# OpenReview forum: "Exploiting Hierarchical Taxonomies in Pretrained Continual Learning"
_ICLR.cc/2025/Conference — ICLR 2025 Conference Withdrawn Submission_

### Official Review · Reviewer_4ENm · 2024-10-31

**Soundness:** 3
**Presentation:** 4
**Contribution:** 3
**Rating:** 5
**Confidence:** 4

**Summary:**

This paper presents a novel approach to address catastrophic forgetting in prompt-based continual learning by introducing a hierarchical label/concept taxonomy and optimal transport-based adjustment. The hierarchical taxonomy organizes classes into a tree-like structure, helping the model focus on distinguishing classes with similar features. The optimal transport-based method further leverages pretrained models to adjust class relationships dynamically, aiming to reduce confusion between overlapping class distributions. The authors demonstrate the effectiveness of their method through experiments on standard datasets like Split CIFAR-100 and Split ImageNet, showing improved performance over existing baseline methods in class incremental learning tasks.

**Strengths:**

1. The paper presents an interesting combination of hierarchical label taxonomy and optimal transport-based adjustments, which is a unique perspective for addressing class separation in continual learning.
2. By employing a hierarchical taxonomy to organize classes into tree-like structures, the paper takes an efficient approach to focus on distinguishing visually and semantically similar classes, potentially enhancing class discrimination.
3. The paper is well-organized, with a logical flow that makes it relatively easy to follow. This makes it accessible to readers and clearly highlights the motivation behind the chosen approach.

**Weaknesses:**

1.  While the method emphasizes separating classes within the same group to reduce overlap in the latent space, the connection to mitigating catastrophic forgetting is not well articulated. Improving class separation within groups does enhance performance, but it’s unclear how this specifically helps prevent forgetting across tasks. If separation is beneficial, why not enforce separation for all class samples rather than limiting it to within-group separation? Moreover, why isn’t separation applied between groups as well, which could further reduce potential class overlap and confusion?

2. Although the weight matrix \Gamma is computed via optimal transport to capture class distribution relationships, it’s unclear how \Gamma adjusts as new tasks and classes are introduced. In class incremental learning, the distribution of classes can change significantly with each new task. If the weight matrix is calculated only at the beginning, does it remain fixed throughout subsequent tasks, or does it get recalculated? Additionally, if the class distributions for new tasks differ markedly from previous ones, should \Gamma be adjusted to reflect these changes? The paper does not clearly address whether or how \Gamma dynamically adapts to accommodate shifting distributions in a continual learning setting.

3. The paper claims, “In particular, after Task 1, we have representations of ‘oak tree’, ‘mouse’, and ‘porcupine’ located in quite separate locations. However, when Task 2 and then Task 3 arrive, the appearance of ‘willow tree’ and ‘pine tree’ makes the latent space become fuller, and ‘oak tree’ no longer maintains the separation from the remaining classes as before and even its representation may be misassigned to other classes, leading to a remarkable drop in performance.” However, there is no visualization or data provided to substantiate this claim. Specifically, there is no clear evidence of how the feature distribution of “oak tree,” “mouse,” and “porcupine” changes before and after new classes are introduced. Including visualizations (such as t-SNE plots) or quantitative data showing this change in feature space would help validate the claim and illustrate the model’s behavior as new tasks are learned.

4. The paper claims that, “this strategy not only alleviates the forgetting of old knowledge—often caused by new classes that are difficult to distinguish from old ones within the same leaf group, but also enables active knowledge transfer between tasks.” However, there is no clear evidence or analysis in the paper demonstrating active knowledge transfer between tasks. Specifically, the paper lacks experimental results, metrics, or examples showing how knowledge learned from one task actively benefits performance on subsequent tasks. This aspect of “active knowledge transfer” should be substantiated with concrete experimental support to validate the claim.

**Questions:**

For detailed questions and suggestions, please refer to the Weaknesses section

---

### Official Review · Reviewer_oez1 · 2024-11-02

**Soundness:** 2
**Presentation:** 2
**Contribution:** 2
**Rating:** 6
**Confidence:** 4

**Summary:**

This paper proposes a novel regularization loss function that encourages models to focus more
on challenging knowledge areas, thereby enhancing overall performance. This paper also demonstrates the importance of organizing and exploiting data meaningfully rather than lumping it together for training.

However, I have some concerns. Please see Weaknesses

**Strengths:**

This paper proposes a novel regularization loss function that encourages models to focus more
on challenging knowledge areas, thereby enhancing overall performance. This paper also demonstrates the importance of organizing and exploiting data meaningfully rather than lumping it together for training. Experimental results show the effectiveness.

**Weaknesses:**

1. In Section Introduction, there are also some methods using mutual learning/continual learning for CL, please add discussions.
such as
[1] Wang, M., Michel, N., Xiao, L. and Yamasaki, T., 2024. Improving Plasticity in Online Continual Learning via Collaborative Learning. In Proceedings of the IEEE/CVF Conference on Computer Vision and Pattern Recognition (pp. 23460-23469).
[2] Michel, N., Wang, M., Xiao, L. and Yamasaki, T., Rethinking Momentum Knowledge Distillation in Online Continual Learning. In Forty-first International Conference on Machine Learning.

2. Please provide some visualizations to show how the training strategy improves model performance.

3. Please provide some visualizations to show how the testing strategy enhances prediction performance.

4. Which GPT version is leveraged for providing expert knowledge?

5. Please provide more details on generating expert knowledge. Such as, how the prompt is designed.

**Questions:**

See Weaknesses

---

### Official Review · Reviewer_Nm3g · 2024-11-03

**Soundness:** 4
**Presentation:** 2
**Contribution:** 3
**Rating:** 5
**Confidence:** 5

**Summary:**

This paper introduces a new perspective to explain the reason for catastrophic forgetting in pretrained-based Continual learning models, which potentially comes from the uncontrolled growth of incoming classes on the latent space. Moreover, by dynamically building label-based hierarchical taxonomies and leveraging the initial behavior of the pre-trained model, they can identify the challenging knowledge areas that further focus, and determine how to learn them during the sequence of tasks. Evaluation of several benchmarks verifies the effectiveness of the proposed method.

**Strengths:**

-The hierarchical taxonomy is a novel and interesting topic for continual learning because it can connect old and new classes, improving stability and plasticity.

-Evaluation shows the effectiveness of the proposed method.

**Weaknesses:**

The algorithm's description is a bit confusing, and the logical structure is not easy to understand.

-There is no problem with the motivation for improving plasticity and stability through a tree-like taxonomy of concepts/labels. This point is easy for everyone to understand and reasonable. However, the analysis of the motivation of the third paragraph does not seem relevant to the study's content.

**Questions:**

1、L150：does the ”with task accuracy” denote the accuracy of current dataset D_t with the task-specific classifier s_t? Moreover, Figure 2(b) shows that the corresponding accuracy stays almost unchanged, meaning that a learned task-related classifier on the frozen latent space is enough to distinguish the classes belonging to each task. What is the goal of measuring the “with task accuracy" ?

2、 L194: “While existing CL methods emphasize the importance of representation learning to keep classes distinct, none explicitly acknowledge the overlap between new and old tasks as a source of forgetting.” Is this statement accurate? Moreover, Is it the catastrophic forgetting caused by the overlap between the old and new classes, or the catastrophic forgetting caused by the severe domain/class shifts between the old and new classes?

3、L227: “This observation motivates us to develop a deep learning classifier that learns labels or concepts structured in a hierarchical taxonomy.” L282: “Our approach aims to train the backbone network so that all class representations must be distinct, especially within each leaf group, to achieve maximum distinguishability.”  The above two descriptions seem to be contradictory. Is the goal of this work to learn a classifier with high discrimination or a backbone network with high discrimination ability?

4、L345: what is the motivation of a prompt ensemble strategy? For example, why the prompt P_{i}(i\in[1,t-1]) are used to initialize the prompt of task t?

5、How many nodes are contained in the tree-like taxonomy of concepts/labels? For example, how many leaf groups, and how to construct the tree-like taxonomy?

6、Why is there such a big difference in the Wasserterin distance magnitude between Table 1 and Figure 6?

7、””-》“”

---

### Official Review · Reviewer_Vi1t · 2024-11-03

**Soundness:** 2
**Presentation:** 2
**Contribution:** 2
**Rating:** 5
**Confidence:** 4

**Summary:**

This paper presents a novel approach to tackle the problem of catastrophic forgetting in Prompt-based Continual Learning models by exploiting the relationships between continuously emerging class data. The proposed method is specifically designed based on the observation that connected information follows a hierarchical tree structure. This work also analyzes the hidden connections between nodes. In addition, a novel regularization loss function is introduced to help models manage challenging knowledge areas more effectively. In experiments, the proposed model shows improvement over previous methods by leveraging both the hierarchical tree structure and the new loss function.

**Strengths:**

- The proposed approach demonstrates an improvement in overall performance across multiple benchmark datasets, achieving higher accuracy and lower forgetting scores compared to existing methods.
- The idea of using hierarchical tree structure with label/concept taxonomy is somewhat novel to the problem of continual learning.

**Weaknesses:**

- The paper is poorly written with unclear messages and equations. For instance, the problem is not well elaborated, especially about the concept taxonomy for each task. It was quite difficult to follow the objective and understand the continual learning problem with the hierarchical taxonomy prompts within a task. The equations are also not described well and ambiguous. For instance, in the denominator in Eq. 1, the set X has no clear description on the subscript and superscript. These unclear messages make the statements vague.
- As the problem statement is not clear, the continual learning task needs more description. Are the tasks having memory? What is the modality of memory to be stored in each task?
- The losses and predictions used for training and testing are not clearly explained. What forms the total loss during training? What energy-based function is used in the testing phase? Additionally, why is the "energy" function not applicable during training?
- The paper lacks motivation. Although the regularization loss is derived from the contrastive loss, no direct connection is established between addressing catastrophic forgetting and the use of contrastive loss. Also, there is no citation in the manuscript to support this connection.
- The proposed method depends on strong assumptions tied to a specific hierarchical tree structure that is custom-designed for each individual dataset. This reliance on a tailored structure, while effective for targeted datasets, restricts the model’s ability to generalize and adapt to different concepts or datasets that do not follow the same predefined hierarchy. As a result, the method lacks flexibility when applied to broader contexts or varying data structures.

**Questions:**

Please answer the questions in the weakness section.

---

### Note · Authors · 2024-11-13

I have read and agree with the venue's withdrawal policy on behalf of myself and my co-authors.